# Self vs. Other Raters' Assessment of Emotional Intelligence in Private and Public Hospitals: A Comparative Study

Rateb Jalil Sweis, Sawsan Aldaod, Niveen Mazen Alsayyed * and Lilana Salem Sukkari

Business Administration, University of Jordan, 11942 Aman, Jordan
* Correspondence: niveen83mazen@gmail.com

**Abstract:** This study aims to investigate the levels of emotional intelligence for managers in public and private hospitals in Jordan for the purpose of identifying the relative practice of emotional intelligence dimensions by managers in each sector. The research will also look into the differences (gaps) in self- and other-assessed emotional intelligence for managers in both public and private hospitals. As such, the theoretical importance of this research lies in its ability to contribute to filling the missing gap in the literature while forming the basis for or being the object of reference for any future research in the field. The researchers adopted a quantitative research design. Data were collected using a 360-degree questionnaire, in which managers' self-assessments, and the assessments of two other raters (supervisors, peers, or subordinates), were used to measure the levels of managers' emotional intelligence in both public and private hospitals. A total of 179 managers and 358 raters participated in our study. The results of the study revealed that differences between managers' self-assessments and others' assessments might be an indicator of inflated managers' self-assessments. Differences between managers' self-assessments and others' assessments were larger in public hospitals compared with those in private hospitals. Hence, our study provides valuable recommendations and implications to enhance the practice of emotional intelligence among managers both in public and private hospitals in Jordan.

**Keywords:** emotional intelligence; private hospitals; public hospitals; self-assessment; raters' assessments

## 1. Introduction

Numerous studies have demonstrated that emotional intelligence (EI) can influence employee attitudes and behaviors in occupations with emotional demands by boosting job satisfaction and lowering job stress (Sanchez-Gomez et al. 2021).

Emotional intelligence (EI) is described as the intentional and intelligent use of emotions to guide thinking and behavior (Haavisto et al. 2019). Emotional intelligence involves spheres such as stress management, interpersonal and intrapersonal competencies, general mood, and adaptability (Bibi et al. 2016, 2020).

Previous research has shown differences among individuals in terms of emotional intelligence, particularly when it comes to providing them with influential information, which in turn would significantly impact their behaviors at both the individual and professional levels. These individual levels comprise wellbeing—mental, social, and psychological (Mayer et al. 2016; Piqueras et al. 2019; Sanchez-Gomez et al. 2021)—and leadership, job satisfaction, seeking good job opportunities, turnover, and high performance at the professional level (Extremera et al. 2018; Mérida-López et al. 2020; Sharp et al. 2020; Nieto-Flores et al. 2019; Giorgi et al. 2016; Sanchez-Gomez et al. 2021).

Thus, previous research confirmed that emotional intelligence permits managers to apply effective leadership styles (Kotzé and Nel 2017; Sanchez-Gomez et al. 2021). Moreover, researchers in the field have linked personality traits (e.g., extraversion and introversion)

with emotional intelligence that usually follows the "causal chain of perception, understanding, and emotional regulation" (De Haro et al. 2020, p. 1845).

Mayer et al. (2016) claimed that managers who are high in emotional perception can read others' emotions. When emotionally intelligent managers can accurately realize their employees' emotions and differentiate the motives underlying their feelings, they are able to form authentic social interactions and grow intimate relationships with their employees to facilitate the communication of their vision and values to them. Therefore, managers can employ their emotional intelligence to develop their own thinking and vision so that they are open to various perspectives and ideas that may provoke their deeply held beliefs (Miao et al. 2018).

Moreover, emotional intelligence can be considered a major aspect of the manager's competence. This ability clarifies major differences between average and excellent managers (Minárová et al. 2020). Therefore, managers with low emotional intelligence can be recognized by employees as managers with a personal agenda who do not take into consideration their views and emotions (Mühlfeit and Costi 2016).

Lorincová (2018) claimed that employees in organizations prefer to be involved in decisions, and this, of course, would have an impact on their work. For example, involving employees can positively influence their emotions and, in turn, create a feeling of satisfaction and wellbeing. This will eventually lead to increased motivation and performance. Here, Gilar-Corbí et al. (2018) argue that emotionally intelligent managers know how to create a job environment that is filled with sympathy, wellbeing, and recognition.

Goleman (2017) defines emotional intelligence in terms of self-control and the management of feelings, which should be directed effectively while working with others with a sense of cooperation to accomplish a common objective. Moreover, Goleman stresses that neither genetic nor nurturing factors in early childhood have an influence on the level of emotional intelligence. Hence, emotional intelligence proves to be learned on a long-term basis.

Researchers in the field have linked emotional intelligence with effective leadership and higher organizational performance. This means that effective leadership works through emotional intelligence, which can lead to good job attitudes, organizational commitment, and high performance (Goleman et al. 2013; Goleman 2017; Meisler and Vigoda-Gadot 2014; Alsayyed et al. 2020; Webb et al. 2014).

Mishar and Bangun (2014) define emotional intelligence as individuals' ability to evaluate ideas, then modify and regulate feelings to enhance emotional and intellectual growth. Kassymzhanova and Mun (2013) argue that the components of emotional intelligence include a distinct comprehension of one's own emotions, emotions of others, self-control, motivation, adaptability, empathy, and overcoming stress, in addition to other personal qualities that foster efficiency in a variety of facets of life. Thus, different definitions in the previous literature have emphasized interpersonal skills, the willingness to understand others' feelings, and the ability to control their behaviors as the main pillars of emotional intelligence (Lawani 2016).

## 2. Emotional Intelligence

### 2.1. Emotional Intelligence Models

#### 2.1.1. Ability Models

Ability models are mainly concerned with relatively distinct mental processes for emotional information. Such models focus on cognitive ability and view emotional intelligence as a form of intelligence to process and analyze emotional information (Caruso et al. 2002; Day and Carroll 2004). Based on the literature of emotional intelligence, there are different ability models such as the Workgroup Emotional Intelligence Profile (WEIP) (Jordan et al. 2002), Mayer et al.'s (2004) Model, and the facet-level process model of emotional intelligence and job performance (Joseph and Newman 2010; McCleskey 2014). Later, some of these models have been reviewed, such as Mayer's model (Mayer et al. 2004), to include different important types, such as social and personal intelligence, instead of

solely emphasizing emotional abilities such as perception, assimilation, understanding, and management of emotions (Mayer et al. 2016).

### 2.1.2. Trait Models

Trait models are mainly concerned with the recognition of emotional abilities; that is, the belief in self-acknowledgment in terms of understanding, managing, and expressing our emotions in a particular environment (Andrei et al. 2016; Kanesan and Fauzan 2019). Regarding Petrides and Furnham (2001) this model of emotional intelligence contains four components: wellbeing, including self-confidence, happiness, and optimism; sociability including social competence, assertiveness, and managing the feelings of others; self-control, including stress management, regulation of feelings, and control of impulses; and, emotionality, including emotional perception of self and others, expression of emotions, and empathy (Petrides 2009, 2010). Here, researchers claimed that the trait model of emotional intelligence intervenes with the Big Five personality (Petrides 2010; Kanesan and Fauzan 2019).

### 2.1.3. Mixed Models

Mixed models are those models that combine ability concepts and personality traits (Kanesan and Fauzan 2019). Based on Bar-On (1997, 2004) emotional intelligence represents an intersection between both an individual's mental abilities and personal qualities. Bar-On stated that emotional intelligence requires social skills that can facilitate communication, perception, management of individuals' own feelings, and realization and control of other's feelings. In that way, mixed models are mainly concerned with intrapersonal qualities (e.g., self-respect, self-awareness of assertiveness, independence, tenacity), interpersonal skills (e.g., empathy, social responsibility, and human relations), adaptability (e.g., flexibility, problem solving, and proactive approach), stress management (e.g., stress control and tolerance, and patience), and general mood (e.g., happiness, optimism, and positivity) (Bar-On 1997, 2004; Kanesan and Fauzan 2019). In a similar vein, Goleman (2001) stated that models are concerned with self-awareness and self-management (personal) and social awareness and relationship management (social).

### 2.2. Emotional Intelligence in Healthcare

The healthcare sector is rapidly changing due to the availability of different providers for patients' services. This, in turn, would impact customer satisfaction and customer loyalty (Goleman 2017; Al-Hamdan et al. 2017). Thus, the impact of emotional intelligence on individuals' behaviors has become increasingly significant, especially in the healthcare sector, which is characterized by the presence of multidisciplinary teams (Roth et al. 2019). Hence, emotional intelligence has become a key to effective leadership in the healthcare industry. However, it has been given less consideration (Kaiser 2009; Mintz and Stoller 2014; Al-Hamdan et al. 2017).

In the field of public health, emotional intelligence is essential because it affects a practitioner's capacity to interact effectively with patients, demonstrate empathy, cooperate with passion, and encourage sustainable changes of lifestyles in their societies (Johnson 2016).

Furthermore, customer demands and regulatory authority are key to success in healthcare systems. This implies that health organizations strive to maintain an improved quality of service while reducing overall costs at the same time (Mintz and Stoller 2014; Santana López et al. 2022). Hence, managers and leaders need to adopt emotional intelligence as an effective strategy to respond effectively to emerging challenges in the healthcare sector (Snell et al. 2011; Santana López et al. 2022).

Research has shown that the traditional approach adopted by physicians in medical systems to make independent decisions based on medical diagnoses and symptoms contrasts with the effective participatory leadership style that emphasizes a collective decision-making approach (McCleskey 2014; Cope and Murray 2017; Magbity et al. 2020).

Different studies in the health sector have linked high emotional intelligence with positive leadership qualities, lower stress and burnout, and higher job satisfaction. In that sense, emotional intelligence can lead to better patient rapport and patronage (Sharp et al. 2020; Extremera et al. 2018).

Furthermore, emotional intelligence has been studied as a performance indicator and has considered a core element of surgical education and training (Sharp et al. 2020). For example, Tyczkowski et al. (2015) stated that successful nurses are able to improve efficiency and proficiency to release stress and adversity using high emotional intelligence.

Likewise, emotional intelligence has been linked with conflict management, particularly in the nursing context. Hence, educators in nursing are advised to focus more on providing students with the required skills to resolve conflict management and improving emotional intelligence to face the inevitable conflicts that would arise during their work (Chan et al. 2014). For pharmacists, effective practice of such a profession is characterized by the presence of high standards to guarantee patient safety and good health (Hassali et al. 2017). This, of course, needs emotional intelligence training and development for practitioners in the field of pharmacy (Higuchi et al. 2017). Particularly, adaptability and flexibility in the emergent circumstances or during organizational change (Seymour et al. 2017).

In Jordan, there is limited research in the field of emotional intelligence (Al-Hamdan et al. 2017). Therefore, more research on emotional intelligence in different settings is recommended, including the healthcare sector (Tannous and Matar 2010; Mahasneh 2013; Alawneh and Sweis 2016; Al-Hamdan et al. 2017). Our research represents an attempt to bridge this gap by investigating emotional intelligence among different levels—top-, middle-, and lower-level managers in public and private hospitals in Jordan. Moreover, this research investigates differences in emotional intelligence among these levels and between managers in public and private hospitals in Jordan.

Here, different emergent challenges and difficulties facing the healthcare sector in Jordan have caused an increased need for managers with high emotional intelligence, with the ability to adapt to such a changing environment, solve stress, and provide high-quality service (Tyczkowski et al. 2015; Roth et al. 2019).

Hence, this research will shed light on the dimensions of emotional intelligence that require development for managers in both public and private hospitals in Jordan. Considering the fact that the health sector in Jordan lacks previous studies in the field of emotional intelligence and specifically on the level of managers' emotional intelligence (Al-Hamdan et al. 2017; Tannous and Matar 2010; Mahasneh 2013; Alawneh and Sweis 2016), our research comes to bridge this gap in the literature.

### 3. Research Theoretical Model and Hypotheses Development

Researchers used the Emotional Competence Inventory (ECI-2) model, which was developed and reviewed by Boyatzis and Goleman in cooperation with Hay Group education (Hay 2002). The ECI-2 model consists of four emotional intelligence dimensions: self-awareness, self-management, social awareness, and relationship management. The first and second dimensions are concerned with an individual's awareness and others' awareness, while the third and fourth dimensions are concerned with an individual's ability to manage his or her own self and manage his or her relationships with others effectively,

The ECI model is an assessment of emotional intelligence competencies that are primarily focused on superior workplace performance (Cavallo and Brienza 2002; Goleman et al. 2002).

Goleman et al. (2002) defined these competencies as four clusters or dimensions, which are as follows:

1. Self-awareness consists of emotional self-awareness, self-assessment, and self-confidence.
2. Social awareness dimension consists of empathy, service orientation, and organizational awareness.

3.    Self-management dimension consists of emotional self-control, transparency, adaptability, achievement, initiative, and optimism.
4.    Relationship management dimension consists of inspirational leadership, influence, developing others, change catalysts, teamwork and collaboration, and conflict management.

This research will investigate the level of emotional intelligence among managers in private and public hospitals in Jordan. Researchers relied on Goleman's ECI-2 model and its four emotional intelligence dimensions.

ECI-2 is comprised of 72 items. This is a self- and multi-rater instrument that can be used to assess how people perceive emotions in various life and work situations (Boyatzis and Sala 2004). In that way, self-assessment represents an instrument to measure how individual managers recognize their emotional intelligence competencies and how raters evaluate emotional intelligence for those managers. This includes raters' observations in a professional context. Raters can be supervisors, subordinates, or peers. Given that any individual manager's ratings may be skewed, each rater would observe different aspects of emotional intelligence for the same manager. Therefore, a minimum of two raters for each manager is required to reach a level of accuracy (Goleman et al. 2004; Araujo and Taylor 2012). The raters' assessments for each manager are then averaged into a single value of assessment for that manager. The adopted model is shown in Figure 1.

| | SELF | SOCIAL |
|---|---|---|
| **RECOGNITION** | **Self- Awareness**<br>• Emotional Self-Awareness<br>• Self – Assessment<br>• Self - Confidence | **Social Awareness**<br>• Empathy<br>• Service Orientation<br>• Organizational Awareness |
| **REGULATION** | **Self- Management**<br>• Emotional Self- Control<br>• Transparency<br>• Adaptability<br>• Achievement<br>• Initiative<br>• Optimism | **Relationship Management**<br>• Inspirational Leadership<br>• Influence<br>• Developing Others<br>• Change Catalyst<br>• Conflict Management<br>• Teamwork and Collaboration |

**Figure 1.** Emotional Competence Inventory (ECI-2) model (Goleman et al. 2004).

*Research Hypotheses*

The First Research Hypothesis

**H0.1.** *There are no significant differences in the managers' emotional intelligence levels between private and public hospitals in Jordan.*

The four sub-hypotheses derived from this hypothesis are:

**H0.1.1.** *There are no significant differences in the managers' self-awareness levels between private and public hospitals in Jordan.*

**H0.1.2.** *There are no significant differences in the managers' self-management levels between private and public hospitals in Jordan.*

**H0.1.3.** *There are no significant differences in the managers' social awareness levels between private and public hospitals in Jordan.*

**H0.1.4.** *There are no significant differences in the managers' relationship management levels between private and public hospitals in Jordan.*

The Second Research Hypothesis

**H0.2.** *There are no significant differences between self and other raters' assessments of emotional intelligence levels for Jordanian private hospitals' managers.*

The four sub-hypotheses derived from this hypothesis:

**H0.2.1.** *There are no significant differences between self and raters' assessments of self-awareness levels for Jordanian private hospitals' managers.*

**H0.2.2.** *There are no significant differences between self and raters' assessments of self-management levels for Jordanian private hospitals' managers.*

**H0.2.3.** *There are no significant differences between self and raters' assessments of social awareness levels for Jordanian private hospitals' managers.*

**H0.2.4.** *There are no significant differences between self and raters' assessments of relationship management levels for Jordanian private hospitals' managers.*

## 4. Research Methodology

### 4.1. Research Population and Sample

The study population consisted of public and private hospitals that provide healthcare services in Amman, the capital of Jordan. The Ministry of Health operates in Amman with 1873 beds in 5 hospitals, representing 37.8% of all public hospital beds in the country. Private hospitals operate in Amman with 3339 beds in 40 hospitals, representing 75.6% of all private hospital beds in the country (Ministry of Health in Jordan MOH 2018).

The sample included top-, middle-, and lower-level managers in public hospitals with a number of beds equal to or greater than 100. According to the official website of the Ministry of Health (MOH) in Jordan (Ministry of Health in Jordan MOH 2018), managers in large hospitals deal with huge and varied medical and non-medical staff members, as well as patients of different health circumstances and varying demands. Thus, a manager's emotional intelligence is of vital importance in a large hospital. Due to hospitals' internal regulations and privacy policies, no data on the number of managers in hospitals are available to the public; the researcher approached the selected hospitals individually. Moreover, in cooperation with the human resources department in each hospital, the researcher was directed to top-, middle-, and lower-level managers who were conveniently available. Then, those managers recommended that the other two raters fill out the two questionnaires related to the other raters' assessments for the manager, and for this reason, the sample became a convenient one. Two hospitals in the private sector declined to participate in the study due to their internal policies.

Researchers ultimately distributed 246 questionnaires in the public hospitals (with 82 managers and 164 other raters) and 372 questionnaires in the private hospitals (with 124 managers and 248 other raters). The overall response rate for both public and private hospitals was 78.5%.

### 4.2. Data Collection Instrument

Researchers used three questionnaires to assess each individual manager. One questionnaire represents a self-assessment questionnaire, which was filled out by the manager on how frequently he or she practices emotional intelligence competencies at work. Second and third questionnaires were filled out by two other raters (peers, subordinates, or supervisors) who deal with the manager at work on daily basis. Raters filled out questionnaires on how the manager practices emotional intelligence at work.

The questionnaire is divided into two sections: the first section contains the demographic data of the respondents; this section consists of five items (gender, age, educational level, work experience, and job position).

The second section of the questionnaire contains 72 items to study the emotional intelligence variable and its four dimensions: self-awareness, self-management, social awareness, and relationship management.

The items were assessed in the questionnaire using a 5-point Likert scale ranging from 5 to 1.

### 4.3. Validity and Reliability

All items of the instrument were adopted from Goleman et al.'s (2002) Emotional Competence Inventory (ECI), and in order to avoid any issues of misunderstanding, the researcher translated the questionnaire items from English into Arabic; the items were also examined by specialized academics at the University of Jordan—School of Business to further check the translation and the structure of the items.

To evaluate construct validity and confirm the elements of emotional intelligence and its dimensions—self-awareness, self-management, social awareness, and relationship management—and to determine the most appropriate items for each dimension, exploratory factor analysis (EFA) was performed (Sekaran and Bougie 2016).

EFA outcomes exhibited that the KMO statistic for all dimensions was greater than 0.50, and Bartlett's test of sphericity statistics were statistically significant ($p < 0.05$), implying the appropriateness of factor analysis.

For the self-awareness dimension (Table 1), the total variance percentage accumulated in items 1 to 4 is 70.187%. Hence, the data were rearranged into 4 factors: emotional self-awareness, self-assurance, accurate self-awareness: agreeableness, and accurate self-awareness: defensiveness.

**Table 1.** Exploratory factor analysis (EFA) for self-awareness.

| Items | Factor | | | |
|---|---|---|---|---|
| | 1 | 2 | 3 | 4 |
| SA-ESA3 | 0.875 | | | |
| SA-ESA2 | 0.866 | | | |
| SA-ESA1 | 0.852 | | | |
| SA-ESA4 | 0.838 | | | |
| SA-SC1 | | 0.792 | | |
| SA-SC4 | | 0.775 | | |
| SA-SC3 | | 0.740 | | |
| SA-SC2 | | 0.733 | | |
| SA-ASA3 | | | 0.854 | |
| SA-ASA4 | | | 0.809 | |
| SA-ASA1 | | | 0.663 | |
| SA-ASA2 | | | | 0.966 |

Regarding the self-management dimension (Table 2), the total variance percentage accumulated in items 1 to 5 is 66.229%. Hence, the data were rearranged in 5 factors: transparency, achievement, emotional self-control, initiative and self-control, and initiative.

Concerning the social awareness dimension (Table 3), the total variance percentage accumulated in items 1 to 3 is 67.962%. Hence, the data were rearranged in 3 factors: service orientation, empathy, and organizational awareness.

As regards the relationship management dimension (Table 4), the total variance percentage accumulated in items 1 to 4 is 63.300%. Hence, the data were rearranged in 4 factors: inspirational leadership, teamwork and collaboration, conflict management, and change catalyst.

Cronbach's alpha coefficient for the factors resulting from exploratory factor analysis (EFA) range from 0.716 to 0.989, which indicates that all the items are highly consistent and reliable. Table 5 shows the values obtained for Cronbach's alpha coefficient.

**Table 2.** Exploratory factor analysis (EFA) for self-management.

| | Component | | | | |
|---|---|---|---|---|---|
| | **1** | **2** | **3** | **4** | **5** |
| SM-T1 | 0.859 | | | | |
| SM-T4 | 0.856 | | | | |
| SM–O2 | 0.841 | | | | |
| SM-T2 | 0.821 | | | | |
| SM–O1 | 0.697 | | | | |
| SM–O4 | 0.660 | | | | |
| SM–O3 | 0.650 | | | | |
| SM-Ad1 | 0.610 | | | | |
| SM-T3 | 0.590 | | | | |
| SM-Ad2 | 0.489 | | | | |
| SM-I4 | 0.469 | | | | |
| SM-Ad4 | 0.436 | | | | |
| SM-Ac3 | | 0.965 | | | |
| SM-Ac4 | | 0.875 | | | |
| SM-Ac2 | | 0.795 | | | |
| SM-Ac1 | | 0.464 | | | |
| SM-Ad3 | | 0.439 | | | |
| SM-ESC3 | | | 0.899 | | |
| SM-ESC4 | | | 0.692 | | |
| SM-ESC1 | | | 0.486 | | |
| SM-I1 | | | | 0.895 | |
| SM-ESC2 | | | | 0.622 | |
| SM-I2 | | | | | 0.794 |
| SM-I3 | | | | | 0.478 |

**Table 3.** Exploratory factor analysis (EFA) for social awareness.

| | Component | | |
|---|---|---|---|
| | **1** | **2** | **3** |
| ScA-SO2 | 0.980 | | |
| ScA-SO1 | 0.904 | | |
| ScA-SO3 | 0.878 | | |
| ScA-SO4 | 0.782 | | |
| ScA-OA3 | | 0.854 | |
| ScA-E3 | | 0.731 | |
| ScA-E2 | | 0.695 | |
| ScA-E1 | | 0.668 | |
| ScA-E4 | | 0.473 | |
| ScA-OA2 | | | 0.943 |
| ScA-OA1 | | | 0.931 |
| ScA-OA4 | | | 0.608 |

**Table 4.** Exploratory factor analysis (EFA) for relationship management.

| | Component | | | |
|---|---|---|---|---|
| | **1** | **2** | **3** | **4** |
| RM-DO2 | 1.009 | | | |
| RM-DO4 | 0.954 | | | |
| RM-DO3 | 0.933 | | | |
| RM-IL4 | 0.748 | | | |
| RM-IL3 | 0.735 | | | |
| RM-DO1 | 0.735 | | | |
| RM-CC1 | 0.696 | | | |
| RM-CC3 | 0.683 | | | |
| RM-IL1 | 0.671 | | | |
| RM-IL2 | 0.610 | | | |
| RM-I2 | 0.554 | | | |
| RM-TC3 | 0.518 | | | |
| RM-CC4 | 0.498 | | | |
| RM-CM3 | | 0.934 | | |
| RM-CM4 | | 0.744 | | |
| RM-TC4 | | 0.583 | | |
| RM-TC2 | | 0.527 | | |
| RM-I1 | | 0.427 | | |
| RM-I4 | | | 0.857 | |
| RM-I3 | | | 0.686 | |
| RM-CM1 | | | 0.647 | |
| RM-CM2 | | | 0.539 | |
| RM-CC2 | | | | 0.853 |
| RM-TC1 | | | | 0.564 |

**Table 5.** Cronbach's alpha coefficient.

| Study Contrast | Number of Items | Cronbachs' Alpha Coefficient |
|---|---|---|
| **Self-Awareness** | **12** | **0.860** |
| Emotional Self-Awareness | 4 | 0.962 |
| Self-Confidence | 4 | 0.853 |
| Self-Assessment—Agreeableness | 3 | 0.989 |
| Self-Assessment—Defensiveness | 1 | 0.869 |
| **Self-Management** | **24** | **0.898** |
| Self-Management F 1 | 12 | 0.811 |
| Self-Management F 2 | 5 | 0.944 |
| Self-Management F 3 | 3 | 0.926 |
| Self-Management F 4 | 2 | 0.843 |
| Self-Management F 5 | 2 | 0.793 |

**Table 5.** *Cont.*

| Study Contrast | Number of Items | Cronbachs' Alpha Coefficient |
|---|---|---|
| **Social Awareness** | **12** | **0.944** |
| Service Orientation | 4 | 0.716 |
| Empathy | 5 | 0.887 |
| Organizational Awareness | 3 | 0.941 |
| **Relationship Management** | **24** | **0.888** |
| Relationship Management F 1 | 13 | 0.955 |
| Relationship Management F 2 | 5 | 0.876 |
| Relationship Management F 3 | 4 | 0.817 |
| Relationship Management F 4 | 2 | 0.834 |
| **Overall Emotional Intelligence** | **72** | **0.966** |

## 5. Findings

Before conducting the *t*-test for the first hypothesis, Leven's test for homogeneity of variances (the equality of variance) (Gastwirth et al. 2009) was conducted to check if the distribution of the data in the public hospitals was similar in shape to the distribution of the data in the private hospitals, at a level of significance (0.05). The null hypothesis for Leven's test is: H0: Variances of the two groups are equal (i.e., there are no significant differences between the variances of the two groups).

Table 6 shows that the *p*-values for Leven's statistics are more than 0.05 for the emotional intelligence variable as well as for its four dimensions of self-awareness, self-management, social awareness, and relationship management. This indicates that the variances of the two groups (public and private hospitals) are homogeneous.

**Table 6.** Leven's test results for the first hypothesis.

| Variable/Dimension | Private Mean | SD | Public Mean | SD | Leven's Statistic F | Sig. |
|---|---|---|---|---|---|---|
| Self-Awareness | 3.756 | 0.409 | 3.847 | 0.495 | 2.056 | 0.052 |
| Self-Management | 3.482 | 0.414 | 3.550 | 0.349 | 2.942 | 0.087 |
| Social Awareness | 3.889 | 0.453 | 3.944 | 0.532 | 2.678 | 0.069 |
| Relationship Management | 3.417 | 0.411 | 3.590 | 0.383 | 0.068 | 0.795 |
| Emotional Intelligence | 3.636 | 0.342 | 3.733 | 0.349 | 0.379 | 0.539 |

As shown in Table 7, there are two *t*-values obtained for emotional intelligence as well as for each of the four dimensions, the first value assuming equality of variances and the second value assuming inequality of variances. As Leven's test significance values prove equality or homogeneity of variances between public and private hospitals for the emotional intelligence variable and its dimensions, the *t*-values assuming equal variances are adopted.

Emotional Intelligence Variable: Table 7 shows the *t*-test result ($t$ (356) = −2.576, $p = 0.010$), indicating that the *p*-value is less than 0.05 (the level of significance); thus the first main hypothesis (H0.1) is rejected, indicating that there is a significant difference in the managers' emotional intelligence levels between the Jordanian public and private hospitals.

Self-Awareness Dimension: Table 7 shows the t-test result ($t$ (356) = −1.89, $p = 0.060$), indicating that the *p*-value is more than 0.05 (the level of significance); thus the first sub-hypothesis (H0.1.1) is accepted, indicating that there is no significant difference in the managers' self-awareness levels between the Jordanian public and private hospitals.

Self-Management Dimension: Table 7 shows the *t*-test result ($t$ (356) = −1.581, $p = 0.115$), indicating that the *p*-value is more than 0.05 (the level of significance); thus, the sec-

ond sub-hypothesis (H0.1.2) is accepted, indicating that there is no significant difference in the managers' self-management levels between the Jordanian public and private hospitals.

Social Awareness Dimension: Table 7 shows the t-test result ($t$ (356) = −1.043, $p$ = 0.298), indicating that the $p$-value is more than 0.05 (the level of significance); thus, the third sub-hypothesis (H0.1.3) is accepted, indicating that there is no significant difference in the managers' social awareness levels between the Jordanian public and private hospitals.

Relationship Management Dimension: Table 7 shows the t-test result ($t$ (356) = −3.967, $p$ = 0.000), indicating that the $p$-value is less than 0.05 (the level of significance); thus, the fourth sub-hypothesis (H0.1.4) is rejected, indicating that there is a significant difference in the managers' relationship management levels between the Jordanian public and private hospitals.

**Table 7.** *t*-test Results for the first hypothesis.

| | | $t$ | Sig. $p$-Value (2-Tailed) | df | Mean Difference * |
|---|---|---|---|---|---|
| | Equal variances assumed | −2.576 | 0.010 | 356 | −0.097 |
| | Equal variances not assumed | −2.565 | 0.011 | 276.285 | −0.097 |
| **Self-Awareness** | Equal variances assumed | −1.89 | 0.060 | 356 | −0.091 |
| | Equal variances not assumed | −1.802 | 0.073 | 240.076 | −0.091 |
| **Self-Management** | Equal variances assumed | −1.581 | 0.115 | 356 | −0.068 |
| | Equal variances not assumed | −1.654 | 0.099 | 319.224 | −0.068 |
| **Social Awareness** | Equal variances assumed | −1.043 | 0.298 | 356 | −0.055 |
| | Equal variances not assumed | −1.002 | 0.317 | 245.664 | −0.055 |
| **Relationship Management** | Equal variances assumed | −3.967 | 0.000 | 356 | −0.174 |
| | Equal variances not assumed | −4.037 | 0.000 | 295.567 | −0.174 |

* The negative values of the mean differences indicate that the means of public hospitals' assessments are higher than the means of private hospitals' assessments.

The second hypothesis, as demonstrated in Table 8, shows that the $p$-values for Leven's statistics are more than 0.05 for emotional intelligence variable as well as for each of its four dimensions of self-awareness, self-management, social awareness, and relationship management. This indicates that the variances of the two groups (self and other raters) in Jordanian private hospitals are homogeneous.

**Table 8.** Leven's test results for the second hypothesis—private hospitals.

| Variable/Dimension | Self/Mean | SD | Raters/Mean | SD | Leven's Statistic F | Sig. |
|---|---|---|---|---|---|---|
| Self-Awareness | 3.814 | 0.387 | 3.697 | 0.424 | 0.022 | 0.883 |
| Self-Management | 3.570 | 0.411 | 3.394 | 0.406 | 0.710 | 0.400 |
| Social Awareness | 3.994 | 0.426 | 3.783 | 0.456 | 0.076 | 0.782 |
| Relationship Management | 3.458 | 0.385 | 3.376 | 0.433 | 0.500 | 0.480 |
| Emotional Intelligence | 3.709 | 0.356 | 3.562 | 0.367 | 1.841 | 0.176 |

Emotional Intelligence Variable: Table 9 shows the $t$-test result ($t$ (222) = 3.270, $p$ = 0.001), indicating that the $p$-value is less than 0.05 (the level of significance); thus, the second main hypothesis (H0.2) is rejected, indicating that there is a significant difference between self and other raters' assessments of the emotional intelligence level for Jordanian private hospitals' managers.

**Table 9.** *t*-test results for the second hypothesis—private hospitals.

| | | *t* | Sig. *p*-Value (2-Tailed) | df | Mean Difference |
|---|---|---|---|---|---|
| Emotional Intelligence | Equal variances assumed | 3.270 | 0.001 | 222 | 0.147 |
| | Equal variances not assumed | 3.270 | 0.001 | 213.487 | 0.147 |
| Self-Awareness | Equal variances assumed | 2.155 | 0.032 | 222 | 0.117 |
| | Equal variances not assumed | 2.155 | 0.032 | 220.218 | 0.117 |
| Self-Management | Equal variances assumed | 3.214 | 0.002 | 222 | 0.176 |
| | Equal variances not assumed | 3.214 | 0.002 | 221.968 | 0.176 |
| Social Awareness | Equal variances assumed | 3.592 | 0.000 | 222 | 0.212 |
| | Equal variances not assumed | 3.592 | 0.000 | 220.99 | 0.212 |
| Relationship Management | Equal variances assumed | 1.498 | 0.136 | 222 | 0.082 |
| | Equal variances not assumed | 1.498 | 0.136 | 218.989 | 0.082 |

Self-Awareness Dimension: Table 9 shows the t-test result ($t$ (222) = 2.155, $p$ = 0.032), indicating that $p$-value is less than 0.05 (the level of significance); thus, the first sub-hypothesis (H0.2.1) is rejected, indicating that there is significant difference between self and other raters' assessments of the self-awareness level for Jordanian private hospitals' managers.

Self-Management Dimension: Table 9 shows the t-test result ($t$ (222) = 3.214, $p$ = 0.002), indicating that $p$-value is less than 0.05 (the level of significance); thus, the second sub- hypothesis (H0.2.2) is rejected, indicating that there is significant difference between self and other raters' assessments of the self-management level for Jordanian private hospitals' managers.

Social Awareness Dimension: Table 9 shows the t-test result ($t$ (132) = 3.592, $p$ = 0.000), indicating that $p$-value is less than 0.05 (the level of significance); thus, the third sub-hypothesis (H0.2.3) is rejected, indicating that there is significant difference between self and other raters' assessments of the social awareness level for Jordanian private hospitals' managers.

Relationship Management Dimension: Table 9 shows the t-test result ($t$ (222) = 1.498, $p$ = 0.136), indicating that $p$-value is more than 0.05 (the level of significance); thus, the fourth sub-hypothesis (H0.2.4) is accepted, indicating that there is no significant difference between self and other raters' assessments of the relationship management level for Jordanian private hospitals' managers.

## 6. Discussion

### 6.1. The Levels of Emotional Intelligence among Managers of Private and Public Hospitals

The primary objective of this study was to identify the level of emotional intelligence among managers in public and private hospitals in Jordan. This included self-awareness, self-management, social awareness, and relationship management dimensions. Emotional intelligence was measured in this study using Goleman's ECI-2 instrument, in which self-assessments and the assessments of two other raters are considered to reveal the levels of emotional intelligence of the selected managers.

Our findings indicate that the level of emotional intelligence among hospital managers in public and private hospitals is comparable. Here, the dimensions of emotional intelligence revealed that self-awareness and social awareness had the highest value, with a relatively high level of practice by managers in private hospitals, while self-management and relationship management had a lower value and were shown to have a relatively average level of practice by managers in the private hospitals.

Moreover, our results show similar values for the level of practice by managers in the public hospitals; that is, a high level of self-awareness and social awareness, and a lower level of practice with regards to self-management and relationship management that has shown to be a relatively average level of practice as well.

These results indicate that managers in both public and private hospitals in Jordan have relatively high levels of self- and social awareness, while they have lower levels of self- and relationship management, i.e., there is a gap between "awareness" and "management" of emotions among the selected managers. These results are consistent with Kaiser's (2009) results, in which physician leaders scored the highest average in the branch of understanding emotions, while they scored lower averages in the branch of managing emotions. A similar gap was found in Cherry's (2011) study, in which analyzing the means for "importance" and "performance" of emotional intelligence competences for medical directors revealed the presence of a gap between both means; this gap indicated that medical directors were "aware" of the importance of controlling and managing their emotions, yet they might have difficulty practicing this control.

The findings are also consistent with the findings of another study by Rubin Pillay (2008), in which public and private hospitals' managers (clinician and non-clinician) rated "people-related skills of teamwork, conflict resolution, delegation, and sharing of information" and "self-management" skills among the most valuable leadership competences that need improvement through health management training programs in order to have more efficient and effective management of public and private hospitals.

The presence of a gap between emotional awareness and management among the selected managers leads us to emphasize the importance of emotional intelligence training and development to improve the management of emotions dimensions. Depending on the notion that emotional intelligence is a learned skill and cumulative capability rather than an innate talent (Goleman 2017), the development of relationship management is dependent on self-management and social awareness as a solid base, each of which in turn requires self-awareness (Goleman 2017).

*6.2. The Differences between Self and Other Raters' Assessments of the Levels of Emotional Intelligence among Managers in the Private Hospitals*

The second objective of this study was to investigate any significant differences between self and raters' assessments of emotional intelligence among the selected managers. Our findings have shown that there was a significant difference at $p = 0.5$ between the means of self and other raters' assessments of the overall emotional intelligence level of the selected managers in the private hospitals.

Investigations revealed that there were significant differences in the assessments of selected managers in private hospitals by self and other raters. These differences appeared to be in the levels of self-awareness, self-management, and social awareness ($p = 0.5$). The difference in the relationship management dimension was not found to be significant. This implies that managers in private hospitals require training and development in relationship management skills from both themselves and other raters' perspectives.

Furthermore, the means of self-assessments were found to be higher than the means of other raters' assessments of the overall emotional intelligence level in all four dimensions. This indicates an overestimation and inflation in the managers' self-assessments. These findings are consistent with other studies (Fleenor et al. 2010; Shih and Susanto 2010; Rodrigues and Madgaonkar 2013; Burckle and Boyatzis 1999) that claimed that relying on self-assessments alone could be misleading. Our findings are also consistent with the recommendations of the technical manual of the Emotional Capacity Inventory (ECI), which is intended to be used in a 360-degree mode. Self-ratings alone may be useful to provide developmental feedback but not to provide valid and reliable measures of emotional intelligence for research purposes. Self-ratings alone are poor predictors of performance (Wolff 2005). However, the overall scores (self and raters' assessments are combined and averaged) indicate leadership strengths and weaknesses in emotional intelligence (Goleman 2006; Araujo and Taylor 2012; Rodrigues and Madgaonkar 2013).

*6.3. The Differences between Self and Other Raters' Assessments of the Levels of Emotional Intelligence among Managers in the Public Hospitals*

The third objective of this study was to investigate any significant differences between self and raters' assessments of emotional intelligence among the selected managers in the public hospitals. Our findings have shown that there was a significant difference at p. 5 between the means of self and other raters' assessments of the overall emotional intelligence level of the selected managers in the private hospitals.

Investigations revealed that there were significant differences in the assessments of selected managers in public hospitals by self and other raters ($p = 0.5$). These differences appeared to be in all four emotional intelligence dimensions (self-awareness, self-management, social awareness, and relationship management).

Moreover, the means of self-assessments were found to be significantly higher than the means of other raters' assessments of the overall emotional intelligence level in all four dimensions. This indicates an overestimation and inflation in the managers' self-assessments. These findings are consistent with other studies (Fleenor et al. 2010; Shih and Susanto 2010; Rodrigues and Madgaonkar 2013; Burckle and Boyatzis 1999) that claimed that relying on self-assessments alone could be misleading. Once again, our findings came to be consistent with the recommendations of the technical manual of the Emotional Capacity Inventory (ECI), which is intended to be used in a 360-degree mode. Self-ratings alone may be useful to provide developmental feedback but not to provide valid and reliable measures of emotional intelligence for research purposes. Self-ratings alone are poor predictors of performance (Wolff 2005). However, the overall scores (self and raters' assessments are combined and averaged) indicate leadership strengths and weaknesses in emotional intelligence (Goleman 2001; Araujo and Taylor 2012; Rodrigues and Madgaonkar 2013).

It is worth mentioning that differences in the means between self and raters' assessments of the overall level of emotional intelligence and in all four dimensions of emotional intelligence were found to be higher for managers in the public hospitals compared to those managers in the private hospitals. Moreover, our findings show that raters' assessments in both the private and public sectors were close. This implies that managers' self-assessments in the public sector were more inflated.

As a result, we can conclude that the inflated self-image caused the inflated self-assessments of managers in public hospitals. This means that those managers see themselves as "high caliber" individuals who have the abilities and qualifications to manage issues of overcrowding and the long waiting lists that public hospitals often experience, taking into consideration the fact that they work under stress with relatively less pay compared to those managers working in private hospitals

Finally, public hospitals serve a large portion of the population with low and middle socioeconomic status. Hence, we would say that public hospitals are often characterized by overcrowding, delays, and long waiting lists. This would, of course, lead to conflicts and sometimes violent acts (Mishra and Dhar 2001; Chan et al. 2014; Hassali et al. 2017; Aldaod et al. 2019). With the fact that other raters, who in reality work closely with their managers in the same environment with same difficulties and challenges as those exist in the public hospitals, see that their managers lack the skills of emotional intelligence. Thus, the gap between self and raters' assessments of emotional intelligence in the public hospitals was found to be larger than the gap that was found in the private hospitals.

## 7. Conclusions

The main objective of this study is to empirically investigate the level of emotional intelligence in top-, middle-, and lower-level managers in public and private hospitals in Jordan. This includes both own and others' (peers, subordinates, or supervisors) assessments of the selected managers. Our study took into account four dimensions of emotional intelligence: self-awareness, self-management, social awareness, and relationship management (Goleman 2001). Furthermore, our research aims to investigate the significant differences in self and raters' assessments of managers' emotional intelligence in Jordan's

public and private sectors. There was a lack of previous studies on the level of managers' emotional intelligence in healthcare in Jordan. Consequently, this gap in the Jordanian literature has been bridged by conducting this study.

*Findings of the Study*

Managers' overall level of emotional intelligence in the private hospitals is relatively average in practice, while their level of emotional intelligence in the public hospitals is relatively high.

The selected managers in both public and private hospitals in Jordan have relatively high levels of self-awareness and social awareness. On the other hand, those managers have lower levels of self-management and relationship management dimensions, with relatively average practice. This indicates the presence of a gap between awareness and management of emotions among managers in the studied hospitals in Jordan.

Our findings revealed a significantly higher level of managers' overall emotional intelligence in public hospitals compared to private hospitals, which specifically appeared in the relationship management dimension in both populations.

Our findings also revealed significant differences between self and rater assessments of managers' emotional intelligence in both private and public hospitals. Moreover, self-assessments scoring higher in all dimensions in both populations may indicate that managers' self-assessments are inflated.

Thus, our findings indicate a strong need for future training and development programs in the field of emotional intelligence and particularly in "managing emotions" for all managers in the Jordanian hospitals.

One interesting finding is that our results revealed that there were statistically significant differences between males and females in the overall emotional intelligence level, and as a result, depressed females showed a lower level of emotional intelligence than males. Furthermore, there were no significant differences between males and females in intrapersonal, interpersonal, or adaptability levels (Tannous and Matar 2010). Moreover, other demographic variables (experience, education, and age) in our study had less impact on the level of emotional intelligence among the selected managers.

## 8. Recommendations and Implications

Considering the sensitivity of the nature of service and care in the health sector, the development of emotional intelligence for the managers of hospitals can be of vital importance. Emotionally intelligent managers can influence employees' behaviors and attitudes when they have the competences that enable them to manage their own internal states and emotions and, moreover, enable them to build and handle relationships among team members and patients in such a manner that can help achieve desirable outcomes.

Our research findings on the level of emotional intelligence in Jordanian hospital managers led us to the following recommendations:

- The need to raise emotional intelligence awareness in Jordan's public and private hospitals through the development of specialized training programs (Aldaod et al. 2019; Al-Hamdan et al. 2017). Such training should be given for both managerial and non-managerial employees in ways that incorporate the concept of emotional intelligence and emphasize the practical aspect of it.
- There is a need to acknowledge that emotional intelligence can be learned and developed through training and practice as well (Daher 2015; Aldaod et al. 2019). This includes developing new criteria for recruiting managers in public and private hospitals in Jordan, considering emotional intelligence dimensions, and knowing that emotional intelligence is linked with superior leadership performance (Goleman 2001, 2017).
- The need to design and customize programs that help exploit more-practiced emotional intelligence dimensions of self-awareness and social awareness. Such dimensions should be considered as a foundation to improve less-practiced dimensions of self-management and relationship management for managers in public and private

hospitals. This implies that emotional intelligence development is an incremental process (Goleman and Cherniss 2001; Cartwright and Pappas 2008; Daher 2015).

- The need to develop a 360-degree feedback process for managers in Jordanian public and private hospitals in which supervisors, peers, and subordinates provide direct reports on performance feedbacks. Thus, in addition to self-perception, managers can receive inputs and insights about how others in their professional context perceive and evaluate them. In this way, the gaps between self and others' assessments of the managers can be minimized.

## 9. Limitations and Future Research

This research has some limitations, as follows:

- Respondents were unfamiliar with emotional intelligence and its relationship with effective leadership and better organizational outcomes.
- Researchers faced some difficulties in distributing the questionnaires due to a lack of published data about the number of managers in Jordanian hospitals. Therefore, researchers were forced to go to each target hospital individually in cooperation with human resources managers in the selected hospitals. Then, the number of the conveniently located managers and raters was determined by the HR managers, while two private hospitals refused to participate due to privacy policies.
- Researchers found few previous studies that investigated the levels of managerial emotional intelligence and differences among managers and other raters in the public and private health sectors in Jordan.

This research suggests the following future studies:

- Conduct similar research in other governorates' hospitals as well as hospitals with less than 100 beds, including more raters for a manager as each rater sees different aspects of the person (Wolff 2005). This can help reveal more interpretable results and more data to evaluate the gap between self and raters' assessments of emotional intelligence.
- Replicate the study with an investigation of the emotional intelligence competencies within the emotional intelligence dimensions to address more areas of deficiency that might be uncovered.
- Understanding factors that influence the emotional intelligence of managers in Jordan's public and private hospitals would be an important topic for future research.
- Replicate the study to reveal more comprehensive results on the overall levels of emotional intelligence for managerial and non-managerial employees in the Jordanian hospitals.

**Author Contributions:** Conceptualization, R.J.S. and S.A.; methodology, R.J.S. and S.A.; software, R.J.S. and S.A.; validation, R.J.S. and S.A.; formal analysis, R.J.S. and S.A.; investigation, R.J.S. and S.A.; resources, R.J.S. and S.A., N.M.A.; data curation, R.J.S. and S.A., N.M.A., and L.S.S.; writing—original draft R.J.S., and L.S.S.; writing—review and editing, N.M.A., visualization, N.M.A.; supervision, R.J.S.; project administration, R.J.S. and N.M.A.; funding acquisition, NA. All authors have read and agreed to the published version of the manuscript.

**Funding:** This research received no external funding.

**Institutional Review Board Statement:** In our research the supervisor Rateb Sweis permitted the student to proceed using the consent form to provide research participants a confidential base as well as protecting their privacy. Also, The University of Jordan guarantees data protection for both researchers and research participants under the umbrella of Scientific Research Ethics. For more information you can visit: https://research.ju.edu.jo/Pages/Ethics.aspx (accessed on 5 December 2022).

**Informed Consent Statement:** Informed consent was obtained from all subjects involved in the study.

**Conflicts of Interest:** The authors declare no conflict of interest.

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
