# Peer review of "Self vs. Other Raters’ Assessment of Emotional Intelligence in Private and Public Hospitals: A Comparative Study"

_admsci, doi:10.3390/admsci12040194_

Round 1

Reviewer 1 Report

Extensive review of English is required.

There seems to be multiple phrases based on prior literature that are strung together without adequately arguing the links between them. Some information have questionable added-value. The lack of a logical thread makes it difficult for readers to follow author's "train of thought"

Research hypotheses are randomly provided without any kind of explanation to support them. Furthermore, why conduct research if authors hypothesize there will be no significant results or differences on any of their research questions? 

How did the researchers ensure an ethical research process? 

Why did the researchers perform an EFA instead of a CFA on a previously validated instrument?

Author Response

Extensive review of English is required - Done

There seems to be multiple phrases based on prior literature that are strung together without adequately arguing the links between them. Some information have questionable added-value. The lack of a logical thread makes it difficult for readers to follow author's "train of thought".  Done.

Research hypotheses are randomly provided without any kind of explanation to support them. Furthermore, why conduct research if authors hypothesize there will be no significant results or differences on any of their research questions?

Research hypotheses are usually stated in the negative form (null hypothesis form) (Sekaran and Bogie, 2016).

The aim of testing hypotheses is to examine if the null hypothesis (H0) will be rejected in favor of the alternative hypothesis (H1). The null hypothesis is considered true unless the statistical techniques prove the opposite. When testing the null hypothesis, the decision rule is: if the significance level or p-value is less than .05, the null hypothesis (H0) will be rejected and the alternative one (H1) will be accepted instead. Otherwise, the null hypothesis (H0) is accepted (i.e. when the p-value is greater than .05)

How did the researchers ensure an ethical research process?

Regarding the University of Jordan regulations - the supervisor of the student in the Business School can decide and give the approval to the student as a researcher.  This is because the type and nature of business research usually do not require conducting any experiments on humans or animals like any other scientific researches which are experimental in nature.  And if so,  the researcher should apply for ethics approval via scientific research deanship website..  

In our research the supervisor Prof. Rateb Sweis permitted the student to proceed using the consent form to provide research participants a confidential base as well as protecting their privacy.   Also, The University of Jordan guarantees data protection for both researchers and research participants under the umbrella of Scientific Research Ethics.  For more information you can visit: https://research.ju.edu.jo/Pages/Ethics.aspx.

Why did the researchers perform an EFA instead of a CFA on a previously validated instrument?

EFA helps us determine what the factor structure looks like according to how participant respond. Exploratory factor analysis is essential to determine underlying constructs for a set of measured variables and is usually done when implementing an already validated and confirmed questionnaire in a different environment or different  setting.  In our case EfA served our purpose  when we utilized  it to  explore the possible underlying factor structure in the Jordanian setting.  CFA , on the other hand, allows the researcher to test the hypothesis that a relationship between the observed variables and their underlying latent construct(s) exists

Reviewer 2 Report

This is interesting research and I feel it assumes importance since it has been done in the healthcare sector, that needs high levels of emotional intelligence. However, I feel some areas need to be improved, and my comments are as follows –

1.       In the beginning of the abstract, add a sentence regarding the research gap that your research is trying to address.

2.       In introduction section use more recent literature on emotional intelligence.

3.       The literature on emotional intelligence is vast. In your paper, I feel the literature review section needs to be expanded and enriched. Please remember that literature review needs to be funneled down to identifying the research gap that your research is trying to address.

4.       A conceptual diagram, showing the relationship between variables and hypotheses is preferred.

5.       Please add a paragraph immediately after the literature review regarding the context of the study. The context should explain why Jordan, why healthcare sector and why a comparative study of public and private hospitals was selected for the study.

6.       In the method section, the participant demographic information is very limited. Please add more information regarding the age, gender distribution etc. of the manager included for the study.

7.       In the discussion section, please relate your findings with more literature on whether similar studies have been done in other contexts and how do their findings match with your study.

8.       You can also see if such studies have been done in sectors other than healthcare and how do their findings compare to your research findings.

9.       How has the demographic aspects of your sample moderated the findings of your study?

10.   In the discussion section, I would like to see the contribution of your study more clearly elucidated.

11.   Please include a summary of your findings, preferably as a table in the conclusion section.

Author Response

  1. In the beginning of the abstract, add a sentence regarding the research gap that your research is trying to address. Done
  2. In introduction section use more recent literature on emotional intelligence. Done
  3. The literature on emotional intelligence is vast. In your paper, I feel the literature review section needs to be expanded and enriched. Please remember that literature review needs to be funneled down to identifying the research gap that your research is trying to address. Done
  4. A conceptual diagram, showing the relationship between variables and hypotheses is preferred. Done
  5. Please add a paragraph immediately after the literature review regarding the context of the study. The context should explain why Jordan, why healthcare sector and why a comparative study of public and private hospitals was selected for the study. Done
  6. In the method section, the participant demographic information is very limited. Please add more information regarding the age, gender distribution etc. of the manager included for the study. Done
  7. In the discussion section, please relate your findings with more literature on whether similar studies have been done in other contexts and how do their findings match with your study. Done
  8. You can also see if such studies have been done in sectors other than healthcare and how do their findings compare to your research findings. Done
  9. How has the demographic aspects of your sample moderated the findings of your study? Done
  10. In the discussion section, I would like to see the contribution of your study more clearly elucidated. Done
  11. Please include a summary of your findings, preferably as a table in the conclusion section. Done. In the form of points not a table.

Round 2

Reviewer 1 Report

Thank you for your responses to my comments. The extensive review has greatly improved your paper. I would recommend another language revision before publication, as there are still many errors in the manuscript (incomplete sentences, grammatical and syntax errors). I also recommend reviewing the references presented in the text so they align with APA guidelines. 

Author Response

Dear Editor,

Hope this email finds you in good health.

Please find attached the manuscript after a second round of language revision.

Looking forward to hearing from you.

Kind Regards
Niveen

Reviewer 2 Report

Thanks for making the necessary changes as per suggestions. Good luck.

Author Response

Dear Editor, 

Thanks for your efforts with us.

Regards

Niveen
